# Identification of Wheat Ideotype under Multiple Abiotic Stresses and Complex Environmental Interplays by Multivariate Analysis Techniques

**DOI:** 10.3390/plants12203540

**Published:** 2023-10-11

**Authors:** Ibrahim Al-Ashkar, Mohammed Sallam, Abdullah Ibrahim, Abdelhalim Ghazy, Nasser Al-Suhaibani, Walid Ben Romdhane, Abdullah Al-Doss

**Affiliations:** Department of Plant Production, College of Food and Agriculture Sciences, King Saud University, Riyadh 11451, Saudi Arabia; msallam@ksu.edu.sa (M.S.); adrahim@ksu.edu.sa (A.I.); aghazy@ksu.edu.sa (A.G.); nsuhaib@ksu.edu.sa (N.A.-S.); wromdhane@ksu.edu.sa (W.B.R.); aaldoss@ksu.edu.sa (A.A.-D.)

**Keywords:** genetic stability, heatmap, ideotype, multiple abiotic stresses, MGIDI, WAASB index

## Abstract

Multiple abiotic stresses negatively impact wheat production all over the world. We need to increase productivity by 60% to provide food security to the world population of 9.6 billion by 2050; it is surely time to develop stress-tolerant genotypes with a thorough comprehension of the genetic basis and the plant’s capacity to tolerate these stresses and complex environmental reactions. To approach these goals, we used multivariate analysis techniques, the additive main effects and multiplicative interaction (AMMI) model for prediction, linear discriminant analysis (LDA) to enhance the reliability of the classification, multi-trait genotype-ideotype distance index (MGIDI) to detect the ideotype, and the weighted average of absolute scores (WAASB) index to recognize genotypes with stability that are highly productive. Six tolerance multi-indices were used to test twenty wheat genotypes grown under multiple abiotic stresses. The AMMI model showed varying differences with performance indices, which disagreed with the trait and genotype differences used. The G01, G12, G16, and G02 were selected as the appropriate and stable genotypes using the MGIDI with the six tolerance multi-indices. The biplot features the genotypes (G01, G03, G11, G16, G17, G18, and G20) that were most stable and had high tolerance across the environments. The pooled analyses (LDA, MGIDI, and WAASB) showed genotype G01 as the most stable candidate. The genotype (G01) is considered a novel genetic resource for improving productivity and stabilizing wheat programs under multiple abiotic stresses. Hence, these techniques, if used in an integrated manner, strongly support the plant breeders in multi-environment trials.

## 1. Introduction

Wheat is a main staple food crop cultivated all over the world covering more than 220 million hectares and satisfying about 20% of daily diet protein necessities [1]. Despite increased wheat areas planted, we still need to increase productivity by 60% to feed the projected population of 9.6 billion globally by 2050 under the adverse effects of climate change [1,2]. Heat and drought are the major abiotic stresses detrimental to plant development of wheat at various stages of growth, leading to major damage and loss of productivity and quality due to a considerable decline in the activities of antioxidant enzymes and photosynthesis [3,4,5]. Plants adapt various tolerance mechanisms and practices that are complementary to each other at the morpho-physio-biochemical and molecular levels in response to abiotic stresses [1,3,6]. Abiotic stresses also promote the production of reactive oxygen species (ROS) that damage various cellular functions such as chlorophyll degradation and lipid peroxidation [7]. A common stress response involves enzymes for the scavenging of ROS (which are main factors in assessing stress tolerance level in plants) including superoxide dismutase, peroxidase, catalase, and ascorbate peroxidase [1,7,8,9]. Thus, improving yield is a major challenge in confronting food insecurity under the adverse effects of climate change, the limited resources available, and the steady increase in the population. Efforts are therefore made to strengthen wheat genotypes through breeding programs targeting agronomic trials to stressed environments and/or simulating them to have more capacity to be adaptable to abiotic stresses and possess high-yielding characteristics [2,10,11,12].

The agronomic trials’ purpose was to examine the impact of factor level/s on plant characteristics to describe, understand, and analyze natural processes under study [13,14,15]. Toward the end of the trial, the scholars often have more columns (one for each trait), which need analysis to come up with and make inferences on the factor (rows) performance. The analysis of data firstly involves ANOVA (analysis of variance) for each studied attribute aiming at hypothesis testing the effects of the factor/s and secondly identifying which factor is significantly different from which other [16,17,18]. The main objective was to determine the superior factor (this is good for one trait), but it was incredibly difficult to rank the factors according to their performance with multiple traits. Scholars with expertise often take into consideration a combination of plant traits that an “ideal” factor (be it treatments and/or cultivars) should provide. For wheat, for example, plant breeders search for cultivars that have early precocity, a high rate of production, and are adapted to changed environments and stress types [19,20]. Breeding improvement strategies for genotypes rely upon an integrative approach across different morpho-physio-biochemical (at the level of leaves and/or the entire plant) traits [21,22,23,24], aimed at providing regular and timely science-based information to support plant scholars in pinpointing the adaptive behavior of plants, especially with environmental stress. So, the combination of most authoritative traits and high-powered computer modeling of multidimensional data is required to gain a deeper understanding of the complicated mechanisms of the relationships between traits [21,25,26,27]. For that purpose, it is strongly suggested to use multivariate techniques to take into consideration the type of correlation between traits. Multivariate techniques such as the coefficients of stepwise multiple linear regression (SMLR) analysis, cluster analysis (CA), principal component analysis (PCA), and LDA have been extensively used in plant trials. Despite SMLR and CA’s importance, one weakness is the collinearity often observed across a range of assessed traits leading to unfavorable results or bias if not handled correctly [15,28,29]. PCA and LDA have been thoroughly used for dimensionality minimization and visual convergence of a two-way table combining treatments and traits [30,31]. Even though all of the above analyses can provide a holistic view of the relationships between traits, treatments’ classification depending on the trait data continues to be a challenge. Hence, novel multivariate techniques are urgently needed to figure out the best ranking of the treatments depending on the multi-trait stability index (MTSI). Olivoto and Nardino [20] proposed the MGIDI, which was designed to select superior genotypes depending on multiple traits and has been successfully used by plant breeders [20,29,32,33,34]. It can distinguish the strengths and weaknesses of the selected superior genotypes depending on multiple traits [20,34]. Therefore, it is a very powerful tool to select the donors’ parents in future hybridization programs to obtain new recombination by integrating all traits into an ideotype. Using the MGIDI in studies evaluating stability can lead to avoiding unnecessary accounts and better strategic decisions, which facilitates making recommendations for excellent cultivars [15,20,29].

Due to variations in the environmental conditions, the genotype performance may vary from strength to weakness and vice versa. This indicates a genotype × environment interplay (GEI) of the crossover type, which means that we need special strategies for improving crops [12,29,35,36]. The genotypes should have stable yield rates through various seasons and abiotic stresses until they are considered satisfactory by farmers [12,37,38]. The GEI impact is critical for plant breeders because of its negative effects (phenotypic value differs from genotypic value), which adversely impact the selection of adequate genotypes for the environment/s (variety for each region/s) [12,39]. Hence, the use of stability and adaptability analyses is great for selecting preferred genotypes (ideotypes) in multi-environment/multiplicative trials with the AMMI model [12,40,41]. This model combines PCA and ANOVA in a single analysis [12,21,34,42]. The AMMI model has been used extensively in multi-site trial (MST) analysis as it gives a more precise estimate of the GEI with appealing biplot tools [38]. However, it is inefficient when analyzing the structure of the linear mixed-effect model (LMM). For this reason, Olivoto et al. [35] suggested a novel model called the WAASB. The WAASB results from the singular value decomposition of the BLUP (best linear unbiased prediction) matrix for GEI effects generated by an LMM to describe ideal genotypes that bring together stability and high performance [35]. This model joins the distinct attributes of the AMMI and BLUP models into a unique model, which helps plant scholars choose the best genotypes (stable and high performance) in many crops [14,34,43,44]. The BLUP improves the accuracy of the prediction and gives credible estimates of random effects [35,45,46]; even though these two are statistically different, they possess the ability to differentiate the GEI pattern from the random error. The WAASB biplot quantifies the superior genotypes (stable and highly productive) with a two-dimensional plot, which takes into account all of the interplay principal component axes (IPCAs) of the model for GEI effects [14,29,34], so the WAASB gives more reliable results. There are many multivariate analysis techniques such as SMLR, CA, LDA, factor analysis (FA), MGIDI, AMMI, WAASB, and biplots that are actively involved in the effective and reliable detection of wheat ideotypes under multiple abiotic stresses and complex interplays [12,20,26,32,35]. The absolute values of evaluated traits do not express the tolerance performance of the genotype under stress conditions. Thus, the use of tolerance indices can give more precise and reliable estimates of these outcomes [29]. Compared with our earlier study, herein we aimed to (i) assess six tolerance multi-indices (drought and heat) to 20 wheat genotypes during three cropping seasons and the effects of the GEI; (ii) to check the validity of categories and the prediction of new cases that have not been assigned categories; and (iii) identify ideotype(s) for the best performing genotypes and stability by MGIDI and WAASB techniques via the six tolerance multi-indices.

## 2. Materials and Methods

### 2.1. Trial Description and Traits Measurement

We evaluated 20 wheat genotypes—14 doubled haploid lines (DHLs) and 6 varieties (Appendix A)—for three winter seasons that were planted under three varying environmental conditions (optimal conditions, drought stress, and heat stress) with a total of nine trials. The nine trials were designed in randomized complete blocks [47] with three replicates. The type of soil, plot area, seedling rate, fertilization rate, weather conditions (Appendix A), treatments, planting dates, and seasons were described in detail in previous studies [12,29]. Five random samples and/or plants (chosen from middle rows to overcome the environment effects) for each genotype/replicate were taken for twelve physio-morphological traits (photosynthesis rate (Pn), transpiration rate (E), stomatal conductance (Gs) canopy temperature (CT), leaf water content (LWC), relative water content (RWC), flag leaf area (FLA), leaf area index (LAI), green leaves area (GLA), polyphenol oxidase (PPO), catalase (CAT), and peroxidase (POD)) and three agronomic traits (plant height (PH), number of kernels (NKS), thousand kernel weight (TKW)), as described in detail in previous studies [12,26]. Five more traits—days to heading (DH) when flowering 50% of plants; days to maturity (DM) when yellowing of peduncles occurs for 50% of plants; grain-filling duration (GFD), the period between DM and DH; number of spikes (NS) in one square meter; and grain yield (GY)—were evaluated from three lines with a length of two meters.

### 2.2. Stress Tolerance Indices

Based on the data from the foregoing traits, the drought (DTI) and heat (THI) stress tolerance indices of each of the above traits were calculated to obtain six environmental indices (E) in Table 1.

### 2.3. Statistical Analysis of Evaluated Data

#### 2.3.1. ANOVA and Plotting Performance

The data were checked by the normality test to ensure the data quality, that the data were free from outliers, and that the data followed a normal distribution as explained by the Shapiro–Wilk test [48]. Bartlett’s test [49] exhibited the homogeneity of the six environments; so, we used joint ANOVA (a set of season and abiotic stress) for 20 traits which were studied and analyzed supposing that genotypes were fixed factors and replications and environments were random factors. According to a formula, the general linear mixed-model ANOVA is as follows: $$Y_{ijk}=µ+G_{i}+E_{j}+{Rk}_{\left(j\right)}+{GE}_{ij}+\alpha_{ijk}$$
where Y*_ijk_* is the 20 genotypes’ phenotypic value *_i_* for the studied trait in the environment *_j_* and block k, µ is the overall mean, G*_i_* is the impact of the 20 genotypes, E*_j_* is the impact of the six environments, Rk*_(j)_* is the impact of three replications, GE*_ij_* is the interaction impact of 20 genotypes with six environments, and α is the residual error supposing that values are normally distributed and independent, with a mean of 0 and a variance of $\sigma^{2}\;$ [35]. Genetic parameters were calculated from the expected mean of squares in ANOVA. The means resulting from the analysis were used for plotting performance. The AMMI function was used to predict the response variable of a two-way table based on the number of multiplicative terms, which regression plots with predicted value lines using singular value decomposition of the matrix of best linear unbiased predictions (BLUPs) were generated by a linear mixed model [35].

#### 2.3.2. Stepwise Regression, Cluster, and Discriminant Analyses

The predictive relationships were analyzed for 19 independent traits across all pooled data for the six environmental indices by SMLR analysis that were used to identify the main traits that contribute to strengthening and developing the intrigued variable (GY). The indices of traits were used in the arithmetic of the cluster analysis and the displayed heatmap drawing of the genetic dissimilarity matrix for genotypes used (Euclidean distance and Ward’s method of agglomeration) across five tolerance categories. The LDA was used to re-check the validity of the genotype categories by analyzing trait indices (as quantitative measures) for the five tolerance categories (as qualitative measures).

#### 2.3.3. MTSI Analyses

The MGIDI was used to categorize the genotypes based on multiple trait values using singular value decomposition of the matrix of BLUPs for the interaction (G × E) effects generated by a linear mixed model to quantify the stability of each genotype [20]. The ideal genotype is calculated based on the ideotype matrix (the Euclidean distance between the scores of the genotypes and the ideal genotypes) for all traits (with a selection intensity of ~20%). The genotype with the lowest (MTSI) value is closer to the ideotype and therefore provides maximum performance and stability to all environments and traits studied. In the METs, each genotype’s stability is estimated by WAASB scores from the singular value decomposition of the matrix of the best linear unbiased predictions’ interaction (G × E) effects generated by incorporating the AMMI and BLUP methods together as proposed by Olivoto et al. [35]. The genotype selection procedure with simultaneous mean performance and stability (for the GY trait) with a weighting between them was implemented by the WAASB index [35]. The used genotypes were categorized into four quadrants by a storage root number–WAASB biplot helping the joint interpretation of stability and mean performance across different environmental stresses.

#### 2.3.4. Statistical Software

Most statistical analyses were conducted using package “metan” in RStudio, R version 4.2.2 (R Core Team 2023), as per Olivoto and Lúcio [50], for multi-environment trial analyses, and the “pheatmap” package was used in dendrogram clustering. The SMLR and LDA analyses were conducted in the XLSTAT package (vers. 2019.1).

## 3. Results

### 3.1. Joint ANOVA and Genetic Parameters

Joint ANOVA for the treatments (environmental indices (E) and genotypes (G)) displayed highly significant differences (*p* < 0.01) for all studied traits, and the interactions were significant for 12 traits and insignificant for 8 traits (DH, RWC, E, POD, Pn, Gs, PPO, and CAT) (Table 2). The variance components have been valued to know the importance of each of them. The phenotypic variation showed values two or three times greater than genetic values for most of the traits. For the broad-sense heritability (H^2^), the calculations that displayed mixed values were equal to or greater than 20.00% in the RWC trait, or less than or equal to 89.40% in the Gs trait. The genotypic accuracy of selection (As) was high (>82.50%) for all traits. The genotypic CVs were higher than that obtained from the residual CVs for most traits, which indicates that the CVs (g/r) ratio was greater than 1, except for some traits (DH, NS, PH, LWC, RWC, NKS, and TKW), which were less than 1 (Table 2). The genotype–environment correlation (rge) displayed values of more than half (>0.5) for four traits, a sign that the genotypic influence was instrumental in the heritability of these traits. The coefficient of determination of the interaction effects (R^2^gei) showed equal values (0.0) for five traits and values of less than half (>0.5) for the other traits. The heritability on the mean basis (h^2^mg) showed values of >0.7 for all traits (Table 2).

### 3.2. Comparison between the Multi-Indices Performance with the Predictable AMMI Model

The AMMI function is used to predict the outcome variable of a two-way table according to the AMMI model, judging by the number of multiplicative terms used, which regression plots were made with predicted value lines. Figure 1 presents the plotting of the genotypes’ performance comparing the studied values of the indices against the predictions of the AMMI model. Ten traits (DM, GFD, NS, PH, FLA, GLA, LWC, NKS, TKW, and GY) showed clear differences between the indices and predicted indices.

The maximum differences were observed in genotypes G05, G15, G17, and G18 in the DM trait; genotypes G03, G05, G15, G17, G18, and G20 in the GFD trait; genotypes G05, G06, G07, G12, and G17 in the NS trait; genotypes G05 and G20 in the PH trait; genotypes G06 and G09 in the FLA trait; genotypes G02, G04, and G12 in the GLA trait; genotypes G04 and G10 in the LWC trait; genotypes G04, G09, G11, and G20 in the NKS trait; genotypes G01, G09, G16, and G19 in the TKW trait; and genotypes G02, G04, G10, G19, and G20 in the GY trait. Four traits (DH, LAI, CT, and RWC) showed minor differences between the indices and predicted indices. The differences were observed in genotypes G01, G04, G06, and G07 in the DH trait; genotypes G02 and G09 in the LAI trait; genotypes G03, G09, and G16 in the CT trait; and genotypes G02, G06, G10, G11, G19, and G20 in the RWC trait. On the contrary, six traits (Pn, E, Gs, PPO, POD, and CAT) did not show any clear distinctions between the indices and predicted indices (Figure 1).

### 3.3. Identification of Multi-Indices Influential in the Yield Tolerance Multi-Index

All dependent multi-indices data were analyzed with independent multi-indices GY using SMLR, to know the influential multi-indices and the ratio of influence in GY multi-index performance (Table 3). The analysis of SMLR showed that only four multi-indices (Pn, CT, TKW, and DH) from the nineteen were directly influential to the GY multi-index (R^2^ was 0.868, *p* < 0.0001, with a noise value of 0.364), and their contribution ratios were 0.545, 0.190, 0.089, and 0.043, respectively (Table 3). So, these four multi-indices could be used as influential selection criteria to create the tolerance of wheat genotypes for multiple stresses (drought and heat). The regression coefficient (b) had the following values: for Pn, b = 0.626 ***, SE = 0.108; for CT, b = 0.795 ***, SE = 0.176; for TKW, b = 0.593 *, SE = 0.250; and for DH, b = −1.599 *, SE = 0.722. According to the equation of the model (GY, Table 3), GY = 0.408 − 1.599 × DH + 0.593 × TKW + 0.795 × CT + 0.626 × Pn, the predicted regression GY value differed from 0.617 (G15) to 0.923 (G20). The values of predicted error and relative error differed from −0.051(G05) to 0.053 (G17) and from −0.071 (G05) to 0.068 (G17), respectively. Evaluation accuracy (%) differed from 92.884 (G05) to 99.900 (G03), with an average value of 96.745 (Table 3).

### 3.4. Cluster Analysis and Linear Discriminant Analysis

The heatmap of the hierarchical clustering genotypes and traits combined with the dendrogram is summarized in Figure 2. The genotypes and traits were clustered using the tolerance multi-indices values, and genotypes were categorized into five ranks for clustering based upon existing differences in the traits. The closely associated wheat genotypes were grouped into row clusters, which point to the clustering pattern and were related to genetic similarity, and each cluster consisted of a different number of genotypes (Figure 2). The HT (highly tolerant) cluster consisted of five genotypes, G02, G03, G05, G07, and G09; the T (tolerant) cluster consisted of five genotypes, G01, G04, G10, G14, and G16; the M (moderately tolerant) cluster consisted of two genotypes, G15 and G18; the S (sensitive) cluster consisted of four genotypes, G06, G08, G19, and G20; and the HS (highly sensitive) cluster consisted of four genotypes, G11, G12, G13, and G17.

And, likewise, all traits were categorized into three column clusters, based on the degrees of similarity between the traits, which differ from one genotype to another. Cluster-1 consisted of four traits (POD, PPO, LAI, and CAT), and they all are non-related to GY. Cluster-2 consisted of fifteen traits (three traits—Pn, TKW, and DH—related to GY; eleven traits—MD, LWC, PH, RWC, FLA, GFD, NKS, Gs, E, GLA, and NS—unrelated to GY; in addition to one trait—GY). Cluster-3 consisted of one trait only (CT), which was completely separated from the rest of the traits, as evidenced by the heatmap color (Figure 2).

The prior and posterior classification of the five (HT, T, M, S, and HS) groups was verified by LDA. In the analysis of all traits studied, compliance in all genotypes was assessed (% correct = 100%), and the membership probability value = 1, indicating full compatibility between the prior and posterior classification (Table 4). But cross-validation showed that compliance was present in only 5 (G01, G08, G11, G13, and G20) genotypes, and 15 genotypes were misclassified (11 of them were transferred to the nearest group). In the case of the analysis of GY and its related traits (Pn, HD, CT, and TKW), there was compliance in 15 genotypes (% correct = 75%) and misclassification in 5 genotypes (G04, G07, G12, G17, and G20) (Table 5). The values of the membership probability (>0.5) proved compatibility for prior and posterior classification, and when it is a value of less than 0.5, it is transferred to the appropriate classification (Table 5). The cross-validation showed compliance in only 6 (G03, G05, G06, G08, G11, and G13) genotypes, and 14 genotypes were misclassified (7 of them were transferred to the nearest group).

### 3.5. MTSI Analyses

#### 3.5.1. Factor Delineation and Selection of Multi-Tolerant Genotypes

PCA showed that the first eight components (eigenvalue > 1) explained 85.90% of the total variation among the 20 studied traits, and as for GY and its related traits (Pn, HD, CT, and TKW), the first two components with eigenvalues > 1 explained 60.20% of the total variation (Table 6). Concerning all studied traits, FA showed that traits DM, GFD, and NS were compiled in FA1; traits Gs and E were compiled in FA2; traits FLA, GLA, and RWC were compiled in FA3; traits LWC and CT were compiled in FA4; traits LAI, DH, and POD were compiled in FA5; traits PH and PPO were compiled in FA6; traits TKW and CAT were compiled in FA7; and traits NKS, Pn, and GY were compiled in FA8. But as for GY and its four related traits, FA showed that traits DH, Pn, and GY traits were compiled in FA1, and traits CT and TKW were compiled in FA2. The MGIDI index was calculated to identify the multi-tolerant genotypes (drought and heat) when considering all studied traits and also GY and its four related traits. As for selection gains, the MGIDI revealed that the number of traits with desired gains was 16 out of 20 traits considering all studied traits and 4 out of 5 traits considering GY and its 4 related traits. These results suggest that MGIDI provided higher total gains of 80.53% and 7.18% for traits that increased and −1.28% and −0.253 for traits that decreased for all studied traits and also GY and its four related traits (Table 7). Among the selected traits, Gs, PPO, and E showed the highest genetic gains (19.60%, 10.60%, and 6.92%, respectively) for all studied traits, and for GY and its four related traits, the GY trait had the highest genetic gains (7.12%). The WAASBY index of the original population (Xo) with all studied traits and with GY and its four related traits varied from 0.561 and 0.754, the lowest one, for the LAI and GY to 1.14, the highest one, for the CT, respectively.

The genotypes selected using the MGIDI were G01, G12, G16, and G02 with all studied traits and they were G10, G11, G08, and G20 with GY and its related traits (Figure 3A,B). G01 and G10 were very close to the cutting point, suggesting that these genotypes could have distinct features. Taking into account the strengths and weaknesses of the selected genotypes when considering all studied traits, FA1 had the highest contribution for G12 and G16. FA2 and FA7 had the highest contribution for G01, G12, and G16. FA3 and FA5 represented the highest contribution for G02, and FA6 and FA8 represented the highest contribution for G12 (Figure 3C). But when considering GY and its related traits, FA1 had the highest contribution for G10, and FA2 had the highest contribution for G08 and G20 (Figure 3D).

#### 3.5.2. Grain Yield Indices Performance and Stability

The ANOVA for AMMI revealed that main effects due to six environmental indices (E), 20 genotypes (G), and GE interaction (GEI) were highly significant (*p* < 0.001) and contributed 10.36%, 28.55%, and 26.38% of the total variation, respectively (Table 8). This analysis moreover divided the GEI sum of squares into five IPCAs and a residual term. IPCA1–IPCA3 were highly significant (*p* < 0.001) and explained 66.99%, 25.39%, and 7.77% of the total variation due to the GEI, respectively. The cumulative variance in the IPCA1 and IPCA2 was 92.38% for the six indices, three (E1, E3, and E5) indices had a positive correlation (the angle among them was <90°), and three indices (E2, E4, and E6) provided the same results (Figure 4).

This indicates that the magnitude of the interaction effects tends to be the same and independent when applying the same abiotic stress (Figure 4). Additionally, negative correlations (the vector angles > 90°) were observed in the E1 with E4 and E6. The WAASB statistic was used for better descriptions to select ideal genotypes based on both index performance and stability. For that, a biplot was rendered based on the WAASB and mean index yields in four quadrants for comprehensive interpretation and a joint evaluation of the genotypes/environment (Figure 5). The first quarter included the four (E1, E4, E5, and E6) environments and one genotype (G09). This genotype and four environments showed a lower tolerance index compared with the average tolerance index, and it thus plays the biggest role in GEI. Genotypes G02, G04, G07, G08, and G10 combined with environments E2 and E3 were in the second quadrant. These genotypes have an acceptable performance and the environments play a big role in the GEI. The environments in this quadrant provide an above-average tolerance index; so, special adaptations should be investigated within this quadrant, especially the high-tolerance genotypes. The G05, G06, G12, G13, G14, G15, and G19 genotypes located in the third quadrant had widely adapted and lower-than-average tolerances, due to a reduction in WAASB values, suggesting a more stable genotype performance across the environments. The fourth quadrant of the biplot features the genotypes that had low WAASB values and high tolerance. Hence, the G01, G03, G11, G16, G17, G18, and G20 genotypes were identified as the most stable genotypes across the environments (Figure 5). Lower WAASB scores describe the genotypes that have high stability and tolerance. In ascending order of WAASB scores, the seven genotypes G01, G06, G12, G03, G18, G11, and G05 with values 0.032, 0.054, 0.064, 0.075, 0.100, 0.103, and 0.114 were selected as the top genotypes with high stability and tolerance, respectively (Table 9). If we take a closer look at the WAASB results, we find that G01, G03, G06, and G12 are more stable (smaller WAASB values) compared to G02, G07, G08, and G10. This may be because 33.20% of the variance is not being expounded by IPCA1. In our results, only 66.80% of the GEI variance was expounded by IPCA1, and the results showed that G7 had the smallest IPCA1 value (−0.280), so it was more stable when using only the first IPCA (unlike the WAASB result). With regard to the six environments’ stability, the WAASB scores showed the following list in ascending order: E6, E5, E2, E1, E3, and E4.

## 4. Discussion

The plant’s responses to stress are very complex, which are about acclimation, adaptation, and tolerance. These responses vary with the life stage and plant type, and the plant’s continuity of life under stress conditions is linked to its ability to hold the line against stress [29,51,52]. Previous studies stated that most of the productive and morpho-physiological traits in wheat are influenced by the cultivar used, the growing culture, and the interaction between them, which could be interpreted as the growth and development of wheat being regulated by the complex interaction of a lot of factors, such as temperature, light, day length conditions, and land quality [12,21,53,54]. The development of cultivars that combine the best qualities of high productivity and stability under varying abiotic stress (drought and heat) levels is one of the top priorities of plant breeders and the greatest goal in modern breeding programs [6,29]. The GY is influenced by both genetic and environmental factors. So, the multi-environment trials (METs) model the magnificent efforts in modern breeding programs for the valid and reliable selection of genotypes. The reliability of prediction (the expected value is close to the visible value) is crucial for an appropriate genotype recommendation and delimitation of mega-environments [12,15,35]. According to Gauch and Zobel [55], to increase the accuracy of prediction in METs, researchers must utilize statistical models that have high prediction abilities [35]. A special focus on this point was given in our article. Joint ANOVA for the treatments showed highly significant differences for all studied traits, and the interactions were significant for 12 out of 20 traits, suggesting that the genotypes’ tolerance indices differed from one treatment to another (Table 2). Plant breeders rely primarily on the genetic stability of traits. The major advantage of biplots is that all IPCA axes are used, thus allowing GEI not maintained in IPCA1 to be included in the genotypes’ ranking [35].

This model also offers the opportunity to appreciate important parameters of quantitative genetics (*h*^2^, As, CVs (g/r), *rge*, *R*^2^*gei* and *h*^2^*mg*) in greater depth and that these provide material information in a plant breeding program and must be utilized in a MET’s assessment. The phenotypic variation showed values two or three times greater than genetic values for most of the traits, suggesting that the environmental effect played a significant role [12,35]. The *h*^2^ values were highly fluctuating (ranging from 20.00% to 89.40%). But *h*^2^*mg* showed values of >0.7 for all traits, reflecting a significant increase in genetic variation of the genotypes used with an accuracy level of more than 0.82. This high accuracy indicates a high potential for predicting the genetic value [29,35,56]. Correlation estimations are necessary for METs, and the high-value rge refers to a simple interaction (the low value is undesirable for the selection of genotypes) [29,35,51]. The rge showed high values (>0.580) for four out of twenty traits, indicating that the environmental effect played a role in the inheritance of most traits. The genotypic CVs were higher than those obtained from the residual CVs for most traits, which indicates that the CVs (g/r) ratio was greater than 1. These results show small variability within the environment used [2,35]. In the present study, we have shown how the advantages of the AMMI model and the performance of multi-indices may be combined to increase the reliability of MET analysis. A study evaluating rice has shown that the estimates using the AMMI model were closer to the “true” value [57], so predictive precision deserves special attention for model diagnosis in MET analysis [35,58].

When comparing the AMMI model and the performance of multi-indices, we found that ten traits (DM, GFD, NS, PH, FLA, GLA, LWC, NKS, TKW, and GY) showed clear differences, four traits (DH, LAI, CT, and RWC) showed minor differences, and six traits (Pn, Gs, E, PPO, CAT, and POD) did not show any clear differences (Figure 1). These results are explained by the heritability (Table 2). The traits that had high values in heritability showed very low differences in the AMMI model and the performance of multi-indices compared to the traits that had low values in heritability [14,51]. Also, the biplots and AMMI model explained the genotypes’ ranking clearly [35]. SMLR is a meaningful way to comprehend the relationships between influential and affected variables [2,37,52]. We used SMLR to analyze 20 indices studied as influential indices to know which ones are powerful indices of multi-stress tolerance and their contribution to the GY index as an affected index (Table 3). The SMLR results stated that four indices (Pn, CT, TKW, and DH) were influential to the GY index (R^2^ of the SMLR model was 0.868, *p* < 0.0001, with a noise value of 0.364), and their contribution ratios were 0.545, 0.190, 0.089, and 0.043, respectively (Table 3). So, these four multi-indices could be used as influential selection criteria to create the tolerance of wheat genotypes for multiple stresses (drought and heat). It is recognized that the performance of the different genotypes varies from index to index, but at least it depends on one of them [2,53]. Also, some traits might show a positive correlation in the same genotype, whereas others may show a negative correlation. As a result, we noticed that the equation of the model (GY, Table 3), GY = 0.408 − 1.599 × DH + 0.593 × TKW + 0.795 × CT + 0.626 × Pn, showed that the predicted regression GY value, error value, and relative error value differed from 0.617 (G15) to 0.923 (G20), from −0.051 (G05) to 0.053 (G17), and from −0.071 (G05) to 0.068 (G17), respectively, with genotype evaluation accuracy (%) ranging from 92.884 (G05) to 99.900 (G03) with an average value of 96.745 (Table 3) [2,54].

Cluster analysis is proficient when analyzing massive data sets with multiple variables [55,56], and at the same time allows for the grouping of the genotypes with identical traits connected to multi-tolerance. The genotypes were grouped into five (HS, S, M, T, and HT) groups for multi-tolerance by a two-way heatmap clustering pattern using standardized MTI values (Figure 2). The closely associated wheat genotypes were grouped into row clusters, which point to the clustering pattern related to genetic similarity [56]. Each HT and T cluster consisted of five genotypes, the M cluster consisted of two genotypes, and each S and HS cluster consisted of four genotypes. Traits DH, CT, Pn, TKW, and GY played a crucial role in differentiating tolerant and sensitive groups of wheat genotypes (Figure 2); it is a great method and more confident than other traditional assessment metrics [55,56,57]. Although, a lot of researchers have used cluster analysis for classification [12,21,58]. But, cross-validation of the grouping method was not used to enhance the reliability of the classification [2]. The prior and posterior classification of the five (HT, T, M, S, and HT) groups was verified (Table 4 and Table 5). In the analysis of all traits studied, compliance was assessed in all genotypes (% correct = 100%), and in the case of the analysis of GY and its related traits (Pn, HD, CT, and TKW), there was compliance in 15 genotypes (% correct = 75%) and misclassification in 5 genotypes (G04, G07, G12, G17, and G20). But cross-validation showed there was compliance in only 5 (G01, G08, G11, G13, and G20) genotypes, and 15 genotypes were misclassified in the case of the analysis of all traits; additionally, there was compliance in 6 (G03, G05, G06, G08, G11, and G13) genotypes and 14 genotypes were misclassified in the case of the analysis of GY and its related traits (Table 4 and Table 5). Therefore, discriminant analysis can be a useful statistical tool that is accurate and credible in identifying genetic resources [2,22,59].

The genotypes’ ranking depending on multiple traits in conjunction with the view of strengths and weaknesses is a powerful tool that can be used to direct researchers to better recommendations for selecting correct genotypes. In our experiment, we used PCA and FA to pool impactful studied traits (Table 6), and the FA managed to reduce the 20 traits to only 8 factors, which explained 85.9%, and 2 factors for GY and its related traits, which explained 60.20%. Using the MGIDI, Olivoto and Nardino [20] and Olivoto et al. [15] have shown how to select superior and suitable genotypes in plant experiments that combine all selected traits (MTSI) that satisfy the breeders and achieve their desired goals. In our study, the MGIDI was superior in the process of selecting traits with intended gains. It has a higher degree of computational ability, and it has the ability to treat multicollinearity as well as the main advantage, that the requirements are outlined by the breeder before calculating the index [20]. The MGIDI revealed that the number of traits with desired gains was 16 out of 20 traits considering all studied traits and 4 out of 5 traits considering GY and its 4 related traits, and rates of increase were 80.53 and 7.18, respectively (Table 7). Out of 20 genotypes used, 4 wheat genotypes considering all studied traits (G01, G12, G16, and G02) and 4 wheat genotypes considering GY and its related traits (G10, G11, G08, and G20) stand out as desirable genotypes with better mean performance and stability under multi-environment conditions (Figure 3A,B). Compared to our previous analysis using absolute values [29], these results are different, which probably can be due to the use of tolerance indices that give more accurate and reliable estimates of genotype performance under stress conditions. Owing to the importance of the MGIDI in evaluating crop cultivars extensively, it is called speed breeding. The overall goal of plant breeders is to identify multivariate techniques to select suitable genotypes that have productive and consistent performance in varied environmental conditions [15,20,29,34]. So, based on multiple-trait information (MGIDI), the genotypes have been classified as desired and undesired. Assessing the strengths and weaknesses of genotypes serves as a novel technique for better mechanisms of crop management (Figure 3C,D), and the use MTSI and MGIDI in future studies will lead to avoiding unnecessary calculations and allows for strategic decisions to be made more readily [29,60]. The MGIDI ranks the factors into factors that contribute more (plotted at the center and/or close) or contribute less (plotted towards the figure’s edge), which is used to identify distinct genetic traits of the parents for careful selection in future hybridization programs in order to obtain a new recombination known as the ideotype.

Genotypes with high productivity in various environments are the ultimate objective of plant breeders, and they work to develop genotypes to strengthen stability and stabilize yield [12,61]. The ANOVA for AMMI revealed that the GEI was highly significant and contributed 26.38% of the total variation and divided the GEI into five IPCAs, of which IPCA1–IPCA3 were highly significant. These results suggest that the cross-reaction gives rise to an overall response and ranking of the genotypes with the yield indices under different biotic stress conditions (Table 8). In our study, the E1, E3, and E5 indices and the E2, E4, and E6 indices had positive correlations (the angle among them was <90°), indicating that the magnitude of the interaction effects tends to be the same and independent when applying the same abiotic stress (Figure 4). The obtuse angle of vectors in E1 with E4 and E6 points out the negative association between them. A vertical projection from the genotype to the environmental vector detects the extent of the interaction with the environment [12,38,41]. The plot (Figure 4) shows that eleven genotypes (G02, G03, G04, G06, G07, G08, G10, G11, G14, G16, and G17) were unstable across the environments. The novel WAASB model explains the GEI, bringing together the AMMI and BLUP models into a unique index to select genotypes based on both index performance and stability [14,29,34,35]. The WAASB biplot, resulting in four quadrants, was constructed with the storage root number on the *x*-axis and WAASB scores on the *y*-axis (Figure 5). The genotypes that have a high WAASB score compared to the WAASB grand mean were regarded as less stable (Table 9). The WAASB biplot quantifies the stability of genotypes by combining an explanation of the stability and productivity in a two-dimensional plot, taking into account all of the IPCAs of the model for GEI effects not maintained in IPCA1 [14,62], so the WAASB gives more reliable results. When we took a closer look at the WAASB results, we found that G01, G03, G06, and G12 are more stable (smaller WAASB values) compared to G02, G07, G08, and G10. The results showed that G07 and G08 had the smallest IPCA1 values (−0.280 and −0.248, respectively), so they were more stable when using only the first IPCA (unlike the WAASB result). This may be because 66.80% of the GEI variance was expounded by IPCA1 while 33.20% of the variance was not being expounded through it. Eventually, further investigation focused on total comprehension of these modern statistical methods would be of benefit and make the method more consistent and useful, to obtain the best (high productivity and stability) genotypes under environmental stresses and to meet increasing demands for crop wheat because of population growth coupled with extreme climatic variations [12,29,34,63].

## 5. Conclusions

The assessment of plant traits is deemed an important tool for a plant breeder in variation studies and is essential for identifying the plant tolerance of abiotic stresses. Many wheat genotypes showed genetic diversity under abiotic stress conditions. In the present study, the categorization of 20 genotypes of wheat into five tolerance categories was verified by LDA, which indicated that prior and posterior categories were perfectly symmetrical (in the case of the study of all traits) and it varied in 5 genotypes (in the case of the study of the GY trait and related traits), but cross-validation showed variations in both cases. G01, G12, G16, and G02 were selected as the appropriate and stable genotypes using the MGIDI with the six tolerance multi-indices. The biplot features the genotypes G01, G03, G11, G16, G17, G18, and G20 as the most stable and they had high tolerance across the environments. The results of the three analyses (LDA, MGIDI, and WAASB) showed genotype G01 to be the most stable candidate. The genotype G01 is considered a novel genetic resource to optimize productivity and genetic stability in wheat programs under multiple abiotic stresses. Hence, these techniques, if used in an integrated manner, could strongly support the plant breeders in METs.

## Figures and Tables

**Figure 1 plants-12-03540-f001:**
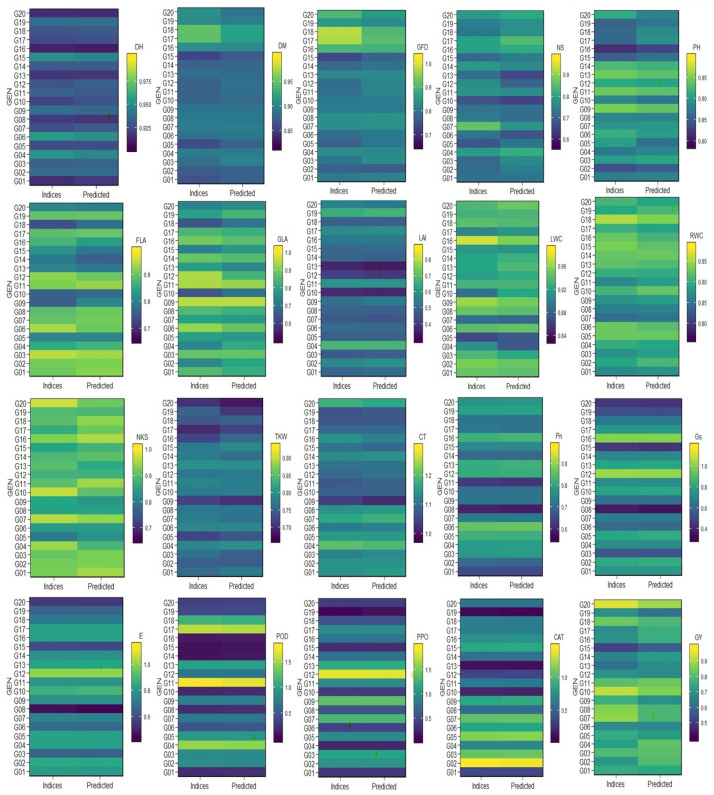
Plotting the multi-indices performance and AMMI model predictions of the 20 traits and genotypes studied as the overall mean during three seasons. Abbreviations are as stated above in the materials and methods section.

**Figure 2 plants-12-03540-f002:**
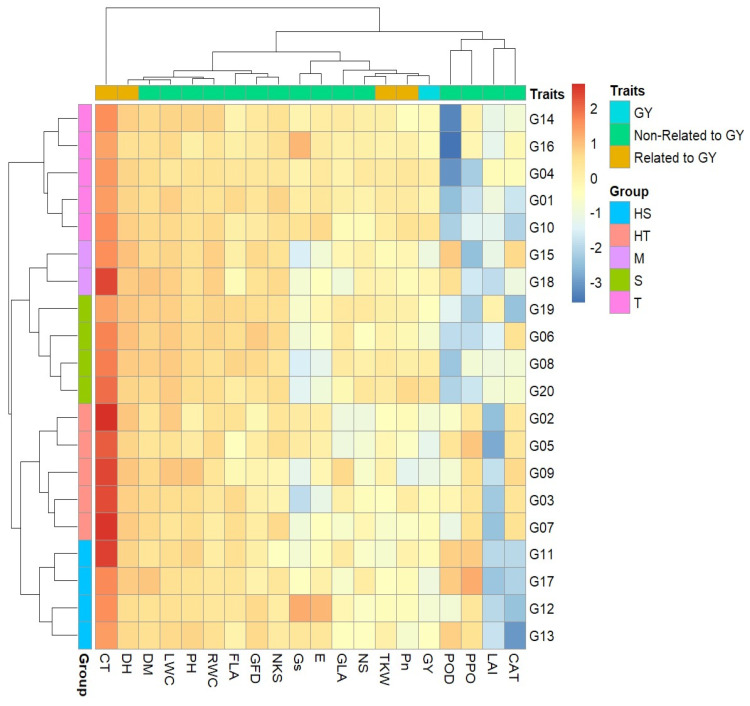
Hierarchical clustering and heatmap of the associations among 20 wheat genotypes dependent on the Euclidean distance with 20 different traits. Colors are representative of a relative scale (−3 to +3) derived from data standardization of the tolerance indices values. HT (highly tolerant), T (tolerant), M (moderately tolerant), S (sensitive), and HS (highly sensitive). Abbreviations are as stated above in the materials and methods section.

**Figure 3 plants-12-03540-f003:**
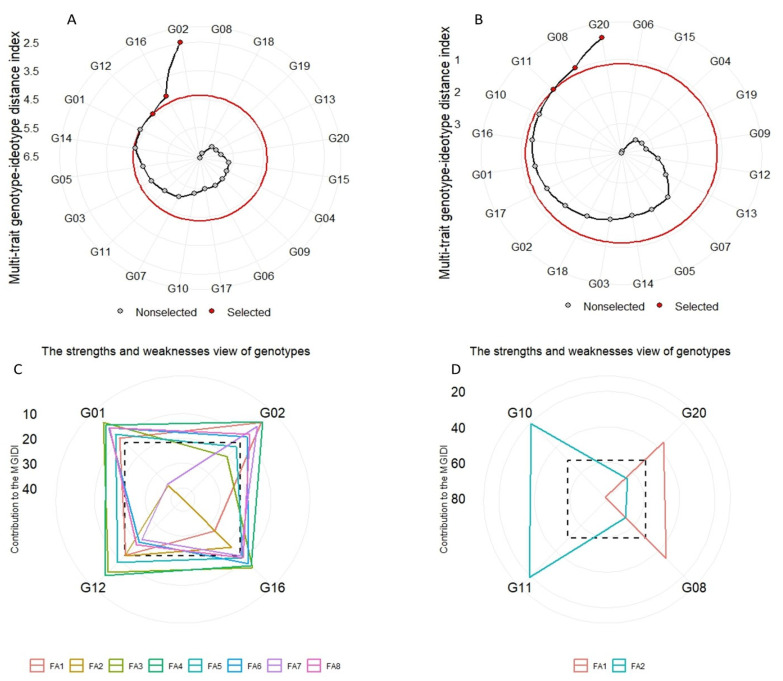
Genotype ranking for the MGIDI with all traits (**A**) and GY and related traits (**B**). The strengths and weaknesses of the selected genotypes with all traits (**C**) and GY and related traits (**D**) are illustrated as the percentage of each factor computed on the basis of MGIDI.

**Figure 4 plants-12-03540-f004:**
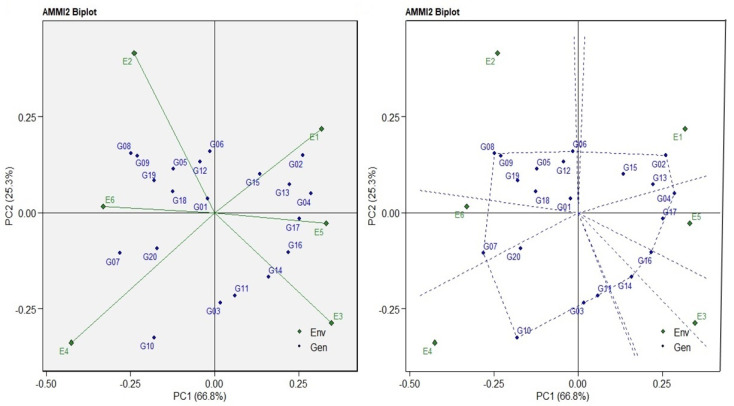
AMMI2 biplot (IPC1 vs. IPC2) for the GY index of 20 wheat genotypes evaluated across six environmental indices.

**Figure 5 plants-12-03540-f005:**
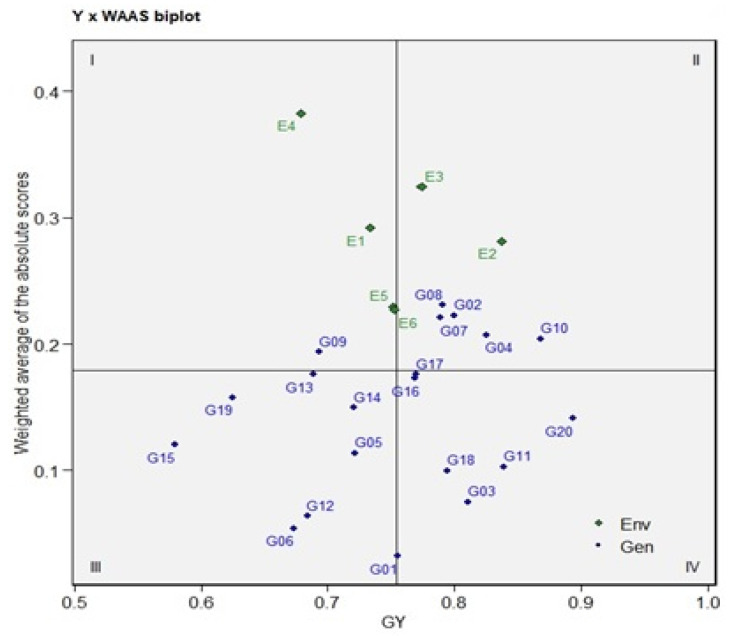
The yield × WAASB biplot based on joint interpretation of storage root number (Y) and stability (WAASB) for 20 wheat genotypes evaluated under six environmental indices.

**Table 1 plants-12-03540-t001:** Environment code and equation of the tolerance indices.

Environment Code	Equation of the Index
E1	DTI*_ij_* = $\frac{x_{ij-under\;drought\;stress\;condition\;in\;season\;2018/19}\;}{x_{ij-under\;optimum\;condition\;in\;season\;2018/19}}$
E2	HTI*_ij_* = $\frac{x_{ij-under\;heat\;stress\;condition\;in\;season\;2018/19}\;}{x_{ij-under\;optimum\;condition\;in\;season\;2018/19}}$
E3	DTI*_ij_* = $\frac{x_{ij-under\;drought\;stress\;condition\;in\;season\;2019/20}\;}{x_{ij-under\;optimum\;condition\;in\;season\;2019/20}}$
E4	HTI*_ij_* = $\frac{x_{ij-under\;heat\;stress\;condition\;in\;season\;2019/20}\;}{x_{ij-under\;optimum\;condition\;in\;season\;2019/20}}$
E5	DTI*_ij_* = $\frac{x_{ij-under\;drought\;stress\;condition\;in\;season\;2020/21}\;}{x_{ij-under\;optimum\;condition\;in\;season\;2020/21}}$
E6	HTI*_ij_* = $\frac{x_{ij-under\;heat\;stress\;condition\;in\;season\;2020/21}\;}{x_{ij-under\;optimum\;condition\;in\;season\;2020/21}}$

DTI*_ij_* and HTI*_ij_* are the drought and heat tolerance indices’ values (j) for genotype (i), respectively.

**Table 2 plants-12-03540-t002:** Joint ANOVA and genetic parameters for 20 studied traits.

Source	Df	DH	DM	GFD	NS	PH	FLA	GLA	LAI	LWC	RWC
ENV	5	0.0422	0.1090	0.2570	0.0995	0.0350	0.0232	0.1740	0.3640	0.0117	0.0107
REP(ENV)	12	0.0006	**0.0003**	**0.0015**	0.0104	**0.0025**	**0.0042**	**0.0021**	**0.0013**	**0.0011**	0.0042
GEN	19	0.0022	0.0036	0.0269	0.0530	0.0147	0.0699	0.0789	0.1350	0.0102	0.0096
GEN:ENV	95	**0.0001**	0.0007	0.0041	0.0138	0.0034	0.0070	0.0144	0.0133	0.0012	**0.0029**
Residuals	228	0.0003	0.0002	0.0013	0.0028	0.0015	0.0025	0.0041	0.0012	0.0009	0.0023
Variance components and genetic parameters											
GEN		0.0002	0.0003	0.0016	0.0029	0.0007	0.0035	0.0046	0.0069	0.0005	0.0006
GEN:ENV		0.0000	0.0002	0.0010	0.0037	0.0006	0.0015	0.0034	0.0041	0.0001	0.0002
Residual		0.0001	0.0001	0.0010	0.0020	0.0010	0.0025	0.0031	0.0011	0.0009	0.0021
Phenotypic variance		0.0004	0.0004	0.0005	0.0036	0.0086	0.0023	0.0075	0.0111	0.0120	0.0015
Heritability		0.6000	0.4906	0.4491	0.3388	0.2959	0.4685	0.4143	0.5730	0.3253	0.2009
R^2^gei		0.0000	0.3620	0.2770	0.4220	0.2270	0.2000	0.3110	0.3380	0.0698	0.0697
h^2^mg		0.8930	0.7930	0.8460	0.7400	0.7720	0.9010	0.8180	0.9010	0.8780	0.7000
Accuracy		0.9450	0.8910	0.9200	0.8600	0.8790	0.9490	0.9040	0.9490	0.9370	0.8370
rge		0.0000	0.5190	0.4340	0.5630	0.2950	0.3750	0.4590	0.7760	0.1030	0.0801
CVg		1.1000	1.3600	4.0300	5.9400	2.7600	6.8600	8.5600	14.700	2.3900	2.1200
CVr		1.6200	1.3300	4.0200	6.7800	4.2300	5.7700	8.0400	6.1000	3.2500	5.2500
CV ratio		0.6790	1.0226	1.0025	0.8761	0.6525	1.1889	1.0647	2.4098	0.7354	0.4038
**Source**	**Df**	**NKS**	**TKW**	**CT**	**Pn**	**Gs**	**E**	**POD**	**PPO**	**CAT**	**GY**
ENV	5	0.1300	0.1450	0.1450	0.1740	0.1610	0.0838	0.0674	0.0558	0.0136	0.1620
REP(ENV)	12	**0.0023**	**0.0018**	**0.0018**	**0.0069**	0.0071	0.0086	0.0093	0.0079	0.0012	**0.0026**
GEN	19	0.0284	0.0191	0.0191	0.0974	0.6010	0.3190	4.3100	3.7400	2.2500	0.1170
GEN:ENV	95	0.0089	0.0030	0.0030	**0.0015**	**0.0030**	**0.0043**	**0.0018**	**0.0016**	**0.0003**	0.0217
Residuals	228	0.0031	0.0021	0.0021	0.0056	0.0027	0.0040	0.0022	0.0026	0.0006	0.0027
Variance components and genetic parameters											
GEN		0.0031	0.0019	0.0019	0.0058	0.0322	0.0175	0.2000	0.1770	0.0950	0.0073
GEN:ENV		0.0020	0.0003	0.0005	0.0000	0.0001	0.0001	0.0000	0.0000	0.0000	0.0063
Residual		0.0011	0.0010	0.0005	0.0044	0.0037	0.0040	0.0421	0.0323	0.0305	0.0007
Phenotypic variance		0.0061	0.0032	0.0029	0.0102	0.0360	0.0216	0.2421	0.2093	0.1255	0.0143
Heritability		0.5057	0.5897	0.6425	0.5674	0.8940	0.8112	0.8262	0.8456	0.7567	0.5091
R^2^gei		0.3220	0.0983	0.1870	0.0000	0.0022	0.0052	0.0000	0.0000	0.0000	0.4410
h^2^mg		0.6860	0.8420	0.9350	0.9550	0.9950	0.9870	1.0000	0.9990	1.0000	0.8150
Accuracy		0.8280	0.9180	0.9670	0.9770	0.9980	0.9930	1.0000	1.0000	1.0000	0.9030
rge		0.3920	0.1350	0.4810	0.0000	0.0275	0.0277	0.0000	0.0000	0.0000	0.7000
CVg		3.7100	3.7300	3.7300	9.0600	24.500	16.700	73.600	56.9000	50.8000	9.6500
CVr		6.2200	5.6500	2.1400	8.3600	7.0300	7.9500	6.8600	6.0000	3.3500	6.9100
CV ratio		0.5965	0.6602	1.7430	1.0837	3.4851	2.1006	10.729	9.4833	15.1642	1.3965

Values in bold indicate insignificance and underline indicates significance at *p* < 0.05.

**Table 3 plants-12-03540-t003:** The SMLR analysis was implemented to create an optimal regression equation (GY).

Source	Coefficient Determination			Model Parameters					
	R^2^ Par.	R^2^ Com.	*p*-Value	Regression Coefficient		Standard Error		*p*-Value	
Intercept				0.408		0.807		0.621	
Pn	0.545	0.545	**<0.0001**	0.626		0.108		**<0.0001**	
CT	0.190	0.735	**<0.0001**	0.795		0.176		**0.000**	
TKW	0.089	0.824	**0.002**	0.593		0.250		**0.032**	
DH	0.043	0.868	**0.001**	−1.599		0.722		**0.043**	
Total R^2^		0.868							
Residual		0.364							
Equation of the model (GY) = 0.408 − 1.599 × DH + 0.593 × TKW + 0.795 × CT + 0.626 × Pn									
Genotypes	Dependent multi-indices				GY	Pred (GY)	Predicted error value	Relative error value	Evaluation accuracy (%)
	DH	TKW	CT	Pn					
G01	0.944	0.821	1.095	0.823	0.755	0.770	−0.015	−0.020	98.013
G02	0.963	0.833	1.137	0.846	0.800	0.794	0.006	0.007	99.270
G03	0.959	0.782	1.172	0.867	0.811	0.810	0.001	0.001	99.900
G04	0.972	0.843	1.150	0.870	0.825	0.811	0.014	0.017	98.278
G05	0.950	0.841	1.115	0.801	0.722	0.773	−0.051	−0.071	92.884
G06	0.975	0.796	1.112	0.763	0.673	0.681	−0.008	−0.012	98.798
G07	0.955	0.749	1.188	0.766	0.789	0.748	0.041	0.052	94.848
G08	0.942	0.816	1.165	0.769	0.790	0.792	−0.001	−0.002	99.810
G09	0.960	0.836	1.162	0.669	0.693	0.709	−0.016	−0.023	97.695
G10	0.952	0.791	1.167	0.877	0.868	0.830	0.038	0.044	95.613
G11	0.955	0.794	1.157	0.871	0.839	0.815	0.024	0.029	97.107
G12	0.957	0.766	1.143	0.764	0.684	0.718	−0.034	−0.050	95.021
G13	0.943	0.802	1.113	0.636	0.689	0.658	0.031	0.046	95.435
G14	0.956	0.808	1.122	0.700	0.720	0.687	0.033	0.046	95.419
G15	0.972	0.727	1.088	0.747	0.579	0.617	−0.038	−0.065	93.502
G16	0.934	0.825	1.097	0.834	0.769	0.796	−0.027	−0.035	96.496
G17	0.951	0.822	1.033	0.836	0.770	0.718	0.053	0.068	93.181
G18	0.956	0.811	1.200	0.776	0.795	0.799	−0.004	−0.005	99.479
G19	0.963	0.752	1.082	0.747	0.625	0.641	−0.016	−0.026	97.413
G20	0.939	0.821	1.207	0.913	0.894	0.923	−0.030	−0.033	96.694
Average									96.745

Values in bold indicate significance.

**Table 4 plants-12-03540-t004:** Posterior probability of membership in heat groupings by LDA with all studied traits.

Genotypes	Classification							Cross-Validation					
	Prior	Posterior	Membership Probabilities					Posterior	Membership Probabilities				
			Pr(HS)	Pr(HT)	Pr(M)	Pr(S)	Pr(T)		HS	HT	M	S	T
G01	T	T	0.000	0.000	0.000	0.000	1.000	T	0.000	0.000	0.000	0.000	1.000
G02	HT	HT	0.000	1.000	0.000	0.000	0.000	**T**	0.000	0.000	0.000	0.000	1.000
G03	HT	HT	0.000	1.000	0.000	0.000	0.000	**M**	0.000	0.000	1.000	0.000	0.000
G04	T	T	0.000	0.000	0.000	0.000	1.000	**M**	0.000	0.000	1.000	0.000	0.000
G05	HT	HT	0.000	1.000	0.000	0.000	0.000	**T**	0.000	0.000	0.000	0.000	1.000
G06	S	S	0.000	0.000	0.000	1.000	0.000	**M**	0.000	0.000	1.000	0.000	0.000
G07	HT	HT	0.000	1.000	0.000	0.000	0.000	**T**	0.000	0.000	0.000	0.000	1.000
G08	S	S	0.000	0.000	0.000	1.000	0.000	S	0.000	0.000	0.000	1.000	0.000
G09	HT	HT	0.000	1.000	0.000	0.000	0.000	**T**	0.000	0.000	0.000	0.000	1.000
G10	T	T	0.000	0.000	0.000	0.000	1.000	**M**	0.000	0.000	1.000	0.000	0.000
G11	HS	HS	1.000	0.000	0.000	0.000	0.000	HS	1.000	0.000	0.000	0.000	0.000
G12	HS	HS	1.000	0.000	0.000	0.000	0.000	**S**	0.000	0.000	0.000	1.000	0.000
G13	HS	HS	1.000	0.000	0.000	0.000	0.000	HS	1.000	0.000	0.000	0.000	0.000
G14	T	T	0.000	0.000	0.000	0.000	1.000	**M**	0.000	0.000	1.000	0.000	0.000
G15	M	M	0.000	0.000	1.000	0.000	0.000	**T**	0.000	0.000	0.000	0.000	1.000
G16	T	T	0.000	0.000	0.000	0.000	1.000	**M**	0.000	0.000	1.000	0.000	0.000
G17	HS	HS	1.000	0.000	0.000	0.000	0.000	**T**	0.000	0.000	0.000	0.000	1.000
G18	M	M	0.000	0.000	1.000	0.000	0.000	**T**	0.000	0.000	0.000	0.000	1.000
G19	S	S	0.000	0.000	0.000	1.000	0.000	**M**	0.000	0.000	1.000	0.000	0.000
G20	S	S	0.000	0.000	0.000	1.000	0.000	S	0.000	0.000	0.000	1.000	0.000

Letters in bold indicate misclassified wheat genotypes. HT (highly tolerant), T (tolerant), M (moderately tolerant), S (sensitive), and HS (highly sensitive).

**Table 5 plants-12-03540-t005:** Posterior probability of membership in heat groupings by LDA with GY and related traits.

Genotypes	Classification							Cross-Validation					
	Prior	Posterior	Membership Probabilities					Posterior	Membership Probabilities				
			Pr(HS)	Pr(HT)	Pr(M)	Pr(S)	Pr(T)		HS	HT	M	S	T
G01	T	T	0.256	0.198	0.000	0.038	0.507	**HS**	0.364	0.270	0.000	0.044	0.322
G02	HT	HT	0.048	0.526	0.000	0.019	0.407	**T**	0.059	0.352	0.000	0.028	0.561
G03	HT	HT	0.098	0.729	0.000	0.010	0.163	HT	0.139	0.610	0.000	0.014	0.236
G04	T	**HT**	0.014	0.662	0.000	0.008	0.316	**HT**	0.001	0.998	0.000	0.000	0.001
G05	HT	HT	0.072	0.819	0.001	0.036	0.072	HT	0.139	0.629	0.005	0.099	0.129
G06	S	S	0.008	0.039	0.000	0.721	0.232	S	0.012	0.058	0.000	0.621	0.310
G07	HT	**T**	0.115	0.201	0.000	0.244	0.440	**T**	0.120	0.037	0.000	0.321	0.522
G08	S	S	0.029	0.007	0.000	0.920	0.044	S	0.135	0.015	0.000	0.743	0.106
G09	HT	HT	0.163	0.826	0.000	0.001	0.011	**HS**	0.635	0.359	0.000	0.000	0.005
G10	T	T	0.024	0.020	0.000	0.407	0.548	**S**	0.027	0.017	0.000	0.667	0.289
G11	HS	HS	0.691	0.244	0.000	0.007	0.058	HS	0.546	0.363	0.000	0.010	0.081
G12	HS	**S**	0.050	0.093	0.027	0.769	0.061	**S**	0.000	0.002	0.001	0.996	0.001
G13	HS	HS	0.955	0.035	0.000	0.001	0.008	HS	0.877	0.109	0.000	0.002	0.012
G14	T	T	0.028	0.046	0.000	0.373	0.553	**S**	0.035	0.054	0.000	0.476	0.435
G15	M	M	0.001	0.002	0.954	0.042	0.001	**S**	0.022	0.023	0.000	0.936	0.018
G16	T	T	0.263	0.102	0.000	0.046	0.589	**HS**	0.614	0.144	0.000	0.056	0.186
G17	HS	**T**	0.410	0.075	0.000	0.003	0.512	**T**	0.000	0.012	0.000	0.000	0.988
G18	M	M	0.000	0.000	1.000	0.000	0.000	**HT**	0.000	0.957	0.042	0.001	0.000
G19	S	S	0.170	0.135	0.081	0.538	0.076	**M**	0.268	0.139	0.584	0.001	0.008
G20	S	**T**	0.006	0.027	0.000	0.415	0.551	**T**	0.003	0.028	0.000	0.020	0.949

Letters in bold indicate misclassified wheat genotypes.

**Table 6 plants-12-03540-t006:** PCA and FA with factorial loadings obtained using varimax rotation and resulting communalities.

All Traits											GY and Related Traits			
Principal Component Analysis (PCA)											Principal Component Analysis (PCA)			
PCA	PC1	PC2	PC3	PC4	PC5	PC6	PC7	PC8	PC9	PC10	PC1	PC2	PC3	PC4
Eigenvalues	3.76	2.81	2.70	2.21	2.06	1.39	1.21	1.03	0.736	0.533	1.78	1.23	0.98	0.61
Variance (%)	18.8	14.1	13.5	11.1	10.3	6.97	6.06	5.13	3.68	2.67	35.70	24.50	19.70	12.10
Cumul (%) *	18.8	32.8	46.4	57.4	67.7	74.7	80.8	85.9	89.6	92.3	35.70	60.20	79.90	92.10
Factor analysis (FA)											Factor analysis (FA)			
Variable	FA1	FA2	FA3	FA4	FA5	FA6	FA7	FA8	Comm ^#^	Uniqu ^$^	FA1	FA2	Comm ^#^	Uniqu ^$^
DM	**0.929**	−0.001	0.012	0.049	−0.072	0.064	0.007	−0.015	0.875	0.125				
GFD	**0.947**	0.010	0.008	0.030	0.227	−0.025	0.075	−0.125	0.971	0.029				
NS	**0.644**	0.063	0.180	−0.150	−0.264	−0.312	−0.407	−0.234	0.861	0.139				
Gs	−0.018	**0.975**	0.036	0.013	−0.114	0.065	−0.053	−0.017	0.972	0.028				
E	0.044	**0.951**	0.080	0.016	0.051	−0.116	−0.074	−0.007	0.934	0.066				
FLA	−0.064	0.063	**−0.884**	0.072	0.177	0.028	−0.045	0.095	0.837	0.163				
GLA	0.083	0.033	**−0.662**	−0.422	0.288	−0.233	0.259	−0.128	0.845	0.155				
RWC	0.071	0.236	**0.800**	−0.159	0.213	0.192	−0.063	−0.265	0.883	0.117				
LWC	0.029	−0.076	−0.076	**−0.859**	−0.243	0.098	−0.040	0.094	0.830	0.170				
CT	0.098	−0.104	−0.262	**0.727**	−0.235	0.399	0.100	0.142	0.862	0.138	−0.324	**−0.780**	0.713	0.287
LAI	−0.264	−0.172	−0.318	−0.084	**0.705**	0.318	0.301	0.086	0.903	0.097				
DH	−0.275	0.070	0.008	−0.160	**−0.782**	0.143	0.011	0.335	0.851	0.149	**−0.750**	0.123	0.578	0.422
POD	−0.400	0.316	−0.090	0.273	**0.454**	−0.521	0.092	0.225	0.880	0.120				
PH	−0.054	0.029	0.137	0.117	−0.012	**0.885**	0.125	0.015	0.835	0.165				
PPO	0.026	0.538	−0.259	−0.003	−0.325	**−0.594**	0.314	−0.117	0.928	0.072				
TKW	0.372	0.243	−0.103	0.360	0.092	−0.033	**−0.678**	0.008	0.807	0.193	−0.333	**0.752**	0.677	0.323
CAT	0.203	0.047	−0.144	0.312	0.192	0.035	**0.753**	0.031	0.767	0.233				
NKS	−0.311	0.250	−0.189	−0.065	−0.041	0.360	−0.035	**0.646**	0.749	0.251				
Pn	−0.162	0.274	0.172	0.179	−0.041	0.161	0.087	**−0.788**	0.819	0.181	**0.558**	−0.033	0.313	0.687
GY	−0.239	0.043	0.010	0.220	−0.188	0.026	0.144	**0.778**	0.769	0.231	**−0.827**	−0.217	0.732	0.268

Values in bold indicate related traits, ^#^ Communality, ^$^ Uniquenesses, and * cumulative variance (%).

**Table 7 plants-12-03540-t007:** Predicted genetic gains for the MGIDI for all traits studied and GY and related traits.

Factor	All Traits				GY and Related Traits			
	VAR	Xo	Sense	MGIDI	VAR	Xo	Sense	MGIDI
FA1	DM	0.927	decrease	−0.03	DH	0.955	decrease	−0.106
FA1	GFD	0.882	decrease	−0.20	Pn	0.794	increase	−5.70
FA1	NS	0.786	decrease	−0.08	GY	0.754	increase	7.12
FA2	Gs	0.744	increase	27.30	TKW	0.802	decrease	−0.147
FA2	E	0.792	increase	13.80	CT	1.14	increase	0.0614
FA3	FLA	0.862	increase	3.09				
FA3	GLA	0.792	increase	0.87				
FA3	RWC	0.91	increase	−0.06				
FA4	LWC	0.935	increase	0.86				
FA4	CT	1.14	increase	0.63				
FA5	DH	0.955	decrease	−0.07				
FA5	LAI	0.561	increase	−0.34				
FA6	PH	0.911	decrease	−0.90				
FA6	POD	0.665	increase	−31.70				
FA6	PPO	0.801	increase	26.80				
FA7	TKW	0.802	increase	0.24				
FA7	CAT	0.695	increase	5.75				
FA8	NKS	0.887	increase	0.88				
FA8	Pn	0.794	increase	0.31				
FA8	GY	0.754	increase	−0.18				
Total (increase)				80.53				7.18
Total (decrease)				−1.28				−0.253

**Table 8 plants-12-03540-t008:** AMMI-ANOVA for the GY index trait for 20 genotypes with six environmental indices.

Source	Df	SS	MS	F Value	Pr (>F)	Proportion	Accumulated
ENV	5	0.809	0.162	62.000	0.0000		
REP (ENV)	12	0.031	0.003	0.962	0.4870		
GEN	19	2.230	0.117	43.200	0.0000		
GEN:ENV	95	2.060	0.022	8.000	0.0000		
IPC1	23	1.380	0.060	22.100	0.0000	66.80	66.80
IPC2	21	0.523	0.025	9.170	0.0000	25.30	92.20
IPC3	19	0.160	0.008	3.100	0.0000	7.70	99.90
IPC4	17	0.001	0.000	0.030	1.0000	0.10	100.00
IPC5	15	0.000	0.000	0.000	1.0000	0.00	100.00
Residuals	228	0.619	0.003				
Total	454	7.810	0.017				

Df (degrees of freedom), SS (sum of squares), MS (mean of squares).

**Table 9 plants-12-03540-t009:** Results for WAASB estimation of 10 wheat genotypes assessed using 6 environmental indices.

Code		Y	IPC1	IPC2	IPC3	IPC4	IPC5	WAAS	
								Value	Rank
Genotypes	G01	0.755	−0.021	0.037	−0.109	−0.006	−0.025	0.032	1
	G02	0.800	0.261	0.149	0.132	−0.011	0.000	0.223	19
	G03	0.811	0.018	−0.233	0.056	−0.018	0.015	0.075	4
	G04	0.825	0.286	0.050	0.039	0.035	−0.019	0.207	17
	G05	0.722	−0.122	0.114	−0.040	−0.015	0.013	0.114	7
	G06	0.673	−0.014	0.160	−0.048	−0.001	0.001	0.054	2
	G07	0.789	−0.280	−0.105	−0.082	0.003	−0.018	0.221	18
	G08	0.790	−0.248	0.154	0.328	0.015	0.011	0.231	20
	G09	0.693	−0.229	0.147	−0.038	−0.012	0.018	0.194	15
	G10	0.868	−0.180	−0.324	−0.021	−0.024	0.007	0.204	16
	G11	0.839	0.060	−0.215	0.106	−0.012	−0.016	0.103	6
	G12	0.684	−0.043	0.132	0.023	0.009	0.009	0.064	3
	G13	0.689	0.222	0.074	−0.108	0.021	−0.008	0.176	13
	G14	0.720	0.160	−0.167	−0.006	0.020	0.055	0.150	10
	G15	0.579	0.135	0.101	−0.057	−0.121	−0.001	0.120	8
	G16	0.769	0.219	−0.103	0.007	0.028	−0.003	0.173	12
	G17	0.770	0.252	−0.016	−0.041	0.015	−0.003	0.176	14
	G18	0.794	−0.125	0.055	0.030	0.013	−0.009	0.100	5
	G19	0.625	−0.179	0.084	−0.214	0.045	0.016	0.158	11
	G20	0.894	−0.171	−0.093	0.041	0.013	−0.042	0.141	9
Environments	E1	0.733	0.318	0.219	0.309	−0.005	0.035	0.292	4
	E2	0.837	−0.239	0.416	−0.208	−0.057	−0.002	0.281	3
	E3	0.774	0.346	−0.286	−0.257	−0.006	0.036	0.324	5
	E4	0.678	−0.426	−0.337	0.160	−0.063	−0.004	0.383	6
	E5	0.751	0.331	−0.028	0.013	0.010	−0.072	0.230	2
	E6	0.753	−0.331	0.017	−0.016	0.121	0.006	0.227	1

## Data Availability

All data is contained within the article or Appendix A.

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
