# Peer review of "Identification of Wheat Ideotype under Multiple Abiotic Stresses and Complex Environmental Interplays by Multivariate Analysis Techniques"

_plants, 2023, doi:10.3390/plants12203540_

Round 1

Reviewer 1 Report

It is a well organised manuscript that fits in the aims of the journal. However, some corrections are needed regarding the materials and methods, references format and the results representation. 

Furthermore, after reading your previous publication: ''Detection of High-Performance Wheat Genotypes and Genetic Stability to Determine Complex Interplay between Genotypes and Environments'' and show the similarity in some points, I would kindly suggest you to add information to the aims of this manuscript as well as in the discussion section in order to discriminate this paper from the previous that you have published.

Please see also comments and corrections in the File uploaded.

Author Response

# Reviewer 1

  • 1- It is a well organised manuscript that fits in the aims of the journal. However, some corrections are needed regarding the materials and methods, references format and the results representation.

Response: Thank you very much for your recommendation, which improves the quality of the manuscript.  We added and modified as required.

  • 2- Furthermore, after reading your previous publication: ''Detection of High-Performance Wheat Genotypes and Genetic Stability to Determine Complex Interplay between Genotypes and Environments'' and show the similarity in some points, I would kindly suggest you to add information to the aims of this manuscript as well as in the discussion section in order to discriminate this paper from the previous that you have published

Response: Thank you very much for your comment. We added the information regarding your great and valuable suggestion in lines 142-145, and lines 589 -591.

  • 3- Please see also comments and corrections in the File uploaded.

Response: Thank you very much for the great and valuable comment and agree with you. We added and modified as required in the red-painted revised version.

Reviewer 2 Report

Dear authors,

Your study aims to address one of the most serious problems nowadays so it is of a great importance. Your findings are meaningful but the text of your manuscript needs an improvement as follows:

1) There is a lack of line numbers so it is difficult for reviewing (especially to point the weaknesses)

2) Please, add an asterisk after the correspondence author (in the line with all authors names below the title)

3) Check the title and the abstract and choose which one is correct - "multiple abiotic stress" or "multiple abiotic stresses"? After that, please, unify in the text.

4) The first sentence in the Introduction section needs to be referred. Please, cite the source of these data.

5) Please, explain why there are some underlined values in Table 1 as you did that for the bolded ones.

6) The font type of numbers in "Table 1. continue" should be corrected

7) Please, check the cluster groups symbols on page 9, first line below Table 2. I think that they should be HT, T, M, S and HS

8) Table 8 and figure 5 should be placed before Discussion section

9) Please, unify the genotypes' abbreviation in the text - is it G01 or G1, G03 or G3, etc.? In the Abstract and Conclusion the G01 was used but in the Results it is presented as G1, and for the others too.

10) The References section do not follow the Instructions for authors requirements. Please, check and correct where needed. 

For instance:

Ref. 1. and 2. The journal's name should be in italic, the year should be in bold

Ref. 3, 9, 11, 13, 14 and others. The journal's name is missing

Ref. 10 and 15. The Latin name of species should be in italic

Author Response

# Reviewer 2

  • Your study aims to address one of the most serious problems nowadays so it is of a great importance. Your findings are meaningful but the text of your manuscript needs an improvement as follows:

Response: Thank you very much for the great and valuable comment and agree. We added the improvements required.

  • 1) There is a lack of line numbers so it is difficult for reviewing (especially to point the weaknesses)

Response: Thank you very much for your comment. We added the line numbers in the revised version.

 2) Please, add an asterisk after the correspondence author (in the line with all authors names below the title)

Response:  We added it.

 3) Check the title and the abstract and choose which one is correct - "multiple abiotic stress" or "multiple abiotic stresses"? After that, please, unify in the text.

Response: Thank you very much for the great comment and agree with you. We added "multiple abiotic stresses" in both.

 4) The first sentence in the Introduction section needs to be referred. Please, cite the source of these data.

Response: Thank you very much for your comment. We added reference [1] in line 41.

  • 5) Please, explain why there are some underlined values in Table 1 as you did that for the bolded ones.

Response: Thank you very much. It was explained below (Table 1).

  • 6) The font type of numbers in "Table 1. continue" should be corrected

Response: It's corrected.

  • 7) Please, check the cluster groups symbols on page 9, first line below Table 2. I think that they should be HT, T, M, S and HS

Response: Thank you very much for your comment. We corrected in line 336.

  • 8) Table 8 and figure 5 should be placed before Discussion section

Response: Thank you very much. We left them for text formatting.

  • 9) Please, unify the genotypes' abbreviation in the text - is it G01 or G1, G03 or G3, etc.? In the Abstract and Conclusion the G01 was used but in the Results it is presented as G1, and for the others too.

Response: Thank you very much for your comment. We have corrected the manuscript.

 10) The References section do not follow the Instructions for authors requirements. Please, check and correct where needed.

Response: Thank you very much for your comment. We have corrected the references section as required.

Round 2

Reviewer 1 Report

Manuscript is ok, only some extra format corrections are mentioned in the file uploaded. I advice you to check also for more format mistakes might exist.

Author Response

MS: plants-2634488

MS Title: Identification of wheat ideotype under multiple abiotic stresses and complex environmental interplays by multivariate analysis techniques

Authors: Ibrahim Al-Ashkar, Mohammed Sallam, Abdullah Ibrahim, Abdelhalim Ghazy, Nasser Al-Suhaibani, Walid Ben Romdhane and Abdullah Al-Doss

Dear Academic Editor of plants

Thank you very much for the valuable comments obtained from the Editor and Reviewers of plants Journal. Kindly, find our response to the comments point by point below. Also, the changes have been made in the MS as colorful or deleted.

# Reviewer 1

-        Manuscript is ok, only some extra format corrections are mentioned in the file uploaded. I advice you to check also for more format mistakes might exist.

Response: Thank you very much for your recommendation, which improves the quality of the manuscript.  We moved Tables 8, 9 Figures 4 and 5 as required. The all format mistakes were corrected (lines 36, 178, 387, 533, 644, 648 and 649).